# Geometrical Scaling Effects in the Mechanical Properties of 3D-Printed Body-Centered Cubic (BCC) Lattice Structures

**DOI:** 10.3390/polym13223967

**Published:** 2021-11-17

**Authors:** Alia Ruzanna Aziz, Jin Zhou, David Thorne, Wesley James Cantwell

**Affiliations:** 1Advanced Materials Research Centre, Technology Innovation Institute, Abu Dhabi 9639, United Arab Emirates; 2School of Mechanical Engineering, Xi’an Jiaotong University, Xi’an 710049, China; jin.zhou@xjtu.edu.cn; 3Aerospace Research and Innovation Center (ARIC), Khalifa University of Science and Technology, Abu Dhabi 127788, United Arab Emirates; david.thorne@ku.ac.ae (D.T.); wesley.cantwell@ku.ac.ae (W.J.C.)

**Keywords:** scaling effects, additive manufacturing, PLA polymer, lattice structures, compression tests, failure modes, finite element

## Abstract

This paper investigates size effects on the mechanical response of additively manufactured lattice structures based on a commercially available polylactic acid (PLA) polymer. Initial attention is focused on investigating geometrical effects in the mechanical properties of simple beams and cubes. Following this, a number of geometrically scaled lattice structures based on the body-centered cubic design were manufactured and tested in order to highlight size effects in their compression properties and failure modes. A finite element analysis was also conducted in order to compare the predicted modes of failure with those observed experimentally. Scaling effects were observed in the compression response of the PLA cubes, with the compression strength increasing by approximately 19% over the range of scale sizes investigated. Similar size-related effects were observed in the flexural samples, where a brittle mode of failure was observed at all scale sizes. Here, the flexural strength increased by approximately 18% when passing from the quarter size sample to its full-scale counterpart. Significant size effects were observed following the compression tests on the scaled lattice structures. Here, the compression strength increased by approximately 60% over the four sample sizes, in spite of the fact that similar failure modes were observed in all samples. Finally, reasonably good agreement was observed between the predicted failure modes and those observed experimentally. However, the FE models tended to over-estimate the mechanical properties of the lattice structures, probably as a result of the fact that the models were assumed to be defect free.

## 1. Introduction

In recent years, there has been an unprecedented increase in the use of additively manufactured parts in a range of engineering applications [1,2,3,4,5]. Additive manufacturing offers engineers new opportunities to produce components of great complexity that were hitherto impossible to manufacture using conventional techniques. Additive manufacturing offers many other advantages, including reduced lead times, on-demand manufacturing, increased supply chain proficiency, shorter times to market and reduced waste [6,7,8,9,10]. Individual build times for complex components can be relatively long, often being measured in many hours or even days. As a result, there is a need to develop robust approaches that can deliver rapid answers regarding the ability to manufacture specific designs for given applications. One way to achieve this is to build smaller components and vary the key parameters in the manufacturing process (such as geometric parameters) to identify how they impact the subsequent printability and performance of the finished part.

Bell and Siegmund [11] investigated size effects in the strength of notched 3D-printed acrylic beams. The authors observed size effects in the strength and fracture toughness of the samples and attributed decreases in mechanical properties in small samples to effects associated with the relative thickness of the print layer. The authors observed a size threshold, below which the samples exhibit a reduction in strength with size. This critical size threshold was estimated to be fifty times the thickness of the print layer. Guessasma et al. [12] studied anisotropic damage development in 3D-printed parts based on an acrylonitrile butadiene styrene (ABS) polymer under compression loading. Size effects in the compressive response of blocks, with sizes ranging from 5 mm to 40 mm, were investigated. They observed slight variations in the elastic region, with no significant differences being noted between the compressed specimens. It was argued that the differences that were observed may be due to the cohesive properties, which were affected by the thermal history, given that the filaments in the larger samples travel a greater distance before the next layer is built. Work by Pagano et al. [13] on the compression behaviour of 3D-printed scaffold-like structures also suggested a similar trend, where the findings revealed a strong specimen size dependence in the elastic modulus.

A number of workers have used scaling techniques to investigate size effects in metals, composite materials and sandwich structures [14,15,16,17,18,19,20,21,22,23,24]. Wen and Jones [14] developed a similitude analysis in order to investigate scaling effects in the low velocity impact response metal plates. Their analysis developed response parameters based on a set of physical input parameters. By adopting the Buckingham π theory, the authors developed twelve independent but non-unique π terms. They showed that the impact force scales according to n^2^ (n is the scale size) and deflection according to n. Wen and Jones [14] then performed a series of drop-weight experiments on mild steel and aluminium alloy flat plates wherein the specimens and test parameters were geometrically scaled. It was found that the experimental data obeyed simple geometrics with the normalized (by n^2^) force and displacement (by n) traces collapsing onto a similar curve for both materials. Experimental studies on the effects of specimen size on the flexural and tensile properties of carbon fibre–epoxy composite beams tested under scaled loading conditions were conducted by Jackson et al. [16,17]. The authors used a similar scaling approach and observed that although the stiffnesses of the scaled composite beams were almost identical, the strength values appeared to be markedly dependent on specimen size. Swanson [20] undertook a series of impact tests using drop-weight and airgun test fixtures on differently sized composite plates and cylinders. The results were found to be in good agreement with the predicted scaling behaviour over the range of parameters studied. Another investigation on the effects of scaling on the strength of notched composite laminates based on a unidirectional carbon-fibre/epoxy prepreg system was conducted by Green et al. [22]. They showed that the strength of the composite decreased with increasing specimen size.

A study on scaling effects in the mechanical response of thermoplastic-based fiber–metal laminates (FMLs) was carried out by Carrillo and Cantwell [23] using a falling weight impact tower at a constant impact velocity. The multilayered structures consisted of stacked aluminum alloy sheets, a self-reinforced polypropylene composite and an interlayer adhesive based on a polypropylene film. The results indicated that simple scaling laws can be applied to predict the dynamic mechanical response of larger structures based on these multi-layered systems. Kashani et al. [24] also studied scaling effects in FMLs subjected to tensile and three-point bending loading conditions. The FML was based on aluminium sheets and a unidirectional glass–epoxy composite. Again, the evidence suggested that the mechanical response of FML structures is relatively insensitive to scale size under both tensile and flexural loading conditions.

Previous studies have provided an understanding of the scalability of load-bearing engineering structures; however, limited research has been carried out to investigate scaling effects in 3D-printed structures. Further work is required to predict the mechanical performance of scaled 3D-printed structures, particularly for energy-absorbing applications [25,26]. In the present study, a combined experimental and numerical program is undertaken to investigate scaling effects in the mechanical properties of 3D-printed lattice structures based on a PLA polymer. Initially, compression and three-point bending tests are conducted to investigate size effects in the load–displacement curves and failure mechanisms. Subsequently, scaled lattice structures are fabricated and subjected to compressive loading. The findings are supported with data from a finite element model that is used to predict the failure modes and mechanical response.

## 2. Materials and Method

In this study, a range of cubes, rectangular bars and lattice structures were manufactured using the fused filament fabrication (FFF) additive manufacturing technique. Here, the BigRep ONE 3D printer (BigRep GmbH, Berlin, Germany) shown in Figure 1 was used to produce the test samples. This BigRep printer is designed for the manufacture of large-scale components and has a maximum build volume of 1005 × 1005 × 1005 mm^3^. A PLA polymer filament with a diameter of 2.85 mm was selected as the printing material to fabricate the samples. A schematic diagram showing the three-dimensional printing procedure is presented in the inset in Figure 1. In this process, a continuous polylactic acid (PLA) polymer filament material is heated and extruded through a nozzle according to the parameters given in Table 1. The majority of the specimens were printed along the horizontal orientation to create nominally fully dense solid lattice structures.

A set of four scaled sizes of solid cube, with an edge length of 20n mm, where n = ¼, ½, ¾ and 1, was initially manufactured, as shown in the schematic presented in Figure 2a and given in Table 2. Following this, rectangular samples were manufactured with length, width and thickness dimensions of 200n, 40n and 8n mm, as shown in Figure 2b and given in Table 2. Finally, four scaled sizes of the BCC lattice structure were fabricated, again corresponding to scale sizes of n = ¼, ½, ¾ and 1, and are as shown in Figure 3a. The BCC lattice structures were modelled as a repetitive unit cell, which included a lower baseplate, using a CAD software package. The CAD models were then converted into a 3D printable format using the Simplify3D software package. The unit cell of the BCC lattice structures with elliptical struts with major and minor axes measured 12.5n mm and 9n mm, respectively. The overall nominal width, depth and height of the test samples were 120n × 120n × 107n mm^3^ as outlined in Table 2
Figure 3a shows photographs of the four scaled sizes of lattice structure. A closer examination of the smaller lattice samples highlights the presence of a number of fine horizontal filaments linked to the manufacturing process. Additional square plates with an edge length of 120n mm and a thickness of 4n mm, i.e., similar to the lower baseplate were produced separately and subsequently bonded to the upper surface of the lattice structures using a two-part Araldite epoxy adhesive. This was necessitated by the fact that it was not possible to build the top plates directly on top of the lattice structures during the manufacturing process. The adhesive was allowed to cure for twenty-four hours before testing.

The effect of build direction on the properties of the lattice structures was investigated by manufacturing an additional set of full-size (n = 1) structures at a ninety-degree angle to the previous build direction, i.e., the continuous polymer filaments therefore extended from the bottom of the sample to the top, rather than from one side of the structure to the other, as was the case in the remaining samples.

Initially, the compression properties of the four scaled sizes of solid cube were investigated. Here, the test samples were placed between circular platens of an Instron 5969 universal testing machine. The cubes were then loaded at a crosshead displacement rate of 4n mm/minute until a crosshead displacement of approximately 11n mm was reached. At this point, the test was stopped and the sample removed.

Scaling effects in the flexural properties of the PLA polymer were also studied. Three-point bending tests were conducted on the aforementioned Instron machine using the flexural test set-up shown schematically in Figure 2b. The flexural samples were placed on two supporting rollers at a predetermined separation. The diameter of the rollers, d, was 20n mm and the support span, S, was set to 180n mm. The crosshead displacement rate was maintained at 8n mm/minute.

The final part of this investigation focused on studying size effects in the BCC lattice structures under compression loading. Here, BCC lattice samples were placed between the previously discussed circular steel platens and loaded at a crosshead displacement rate of 8n mm/minute until the samples were considered to be fully crushed. During the loading process, photographic images of the specimens were captured at predetermined intervals. The load and crosshead displacement values were recorded and subsequently used to compare the performance of the scaled sizes BCC lattice structures.

## 3. Numerical Analysis

Numerical models were developed using the finite element software Abaqus/Explicit to predict the behaviour of the lattice structures following compressive loading. An elasto-viscoplastic constitutive model including post-yield hardening was employed to capture the mechanical behaviour of the polylactic acid (PLA) polymer. Prior to simulating the compressive behaviour of the lattice structures, the compression response of the solid cubes and flexural properties of the PLA polymer was simulated using the developed material constitutive model. The four scaled models (n = ¼, ½, ¾ and 1) for the cubes, three point bending and lattices structures were developed.

### Modelling of the PLA Polymer

An elasto-viscoplastic constitutive model, including post-yield hardening, was employed to predict the response of the PLA polymer. The elasto-viscoplastic model was developed by Mirkhalaf et al. [27] based on the Eindhoven Glassy Polymer (EGP) model proposed by Govaert et al. [28]. Figure 2c shows the elasto-viscoplastic model used to describe the polymer. 

The constitutive model for the PLA polymer is given by:σ^t^ = σ^d^ + σ^h^(1)
where, σ^t^, is the overall stress, σ^d^, is the driving stress and, σ^h^, is the hardening stress associated with the PLA polymer. The driving stress (σ^d^) can be obtained from the viscous component in the model, being based on the elastic deformation during tensile or compressive testing. The driving stress (σ^d^) consists of hydrostatic and deviatoric stress components, whereas the hardening stress (σ^h^) is dependent on the total level of deformation and is fully deviatoric in nature. Here, the total deformation gradient, G, is the product of the elastic deformation gradient, Ge, and the plastic deformation gradient, Gp [29].
G = Ge × Gp(2)

The driving Kirchhoff stress is given by [27]:σ^driving^ = D^e^:ε^e^(3)
in which, D^e^, is the fourth order isotropic elastic tensor [27].
(4)$$D^{e}\equiv 2{\mathrm{GI}}_{s}+\left.\left(K-\frac{2}{3}\right.G\right)\left(I⊗I\right)$$
where the terms, G, and, K, represent the shear and bulk modulus, respectively. In this equation, the symbol, $I_{s},$ represents the fourth order symmetric identity tensor and, I, is the second order identity tensor.

The hardening stress is defined as [27]:σ ^hardening^ = H ε_d_
(5)
where, **ε**_d_, represents the total deviatoric strain and, H, corresponds to the hardening modulus, this being one of the material parameters [29].

The plastic flow rule of the model is given by [27]:(6)$$d^{p}=\frac{s}{{2\eta (\sigma }^{\mathrm{eq}})}$$
where, d^p^, is the spatial plastic stretching tensor and, s, is the deviatoric part of the driving Kirchhoff stress (s = I_d_:σ^d^), with, I_d_, being the fourth order deviatoric identity tensor. The viscosity η(σ^eq^) defined by [29]:(7)$${\eta =A}_{o}\exp\left[\frac{\Delta H}{\mathrm{RT}}+\frac{\mu p}{\tau_{o}}-D_{\infty }+D_{\infty }\exp\left.\left(\frac{-h\sqrt{3}{\bar{\varepsilon }}^{p}}{\sqrt{2}D_{\infty }}\right.\right)\right]\times \left[\sigma^{\mathrm{eq}}/\sin h\left(\frac{\sigma^{\mathrm{eq}}}{\tau_{o}}\right)\right]$$
where, σ^eq^, represents the von Mises equivalent stress and, ${\bar{\varepsilon }}^{p},$ represents the accumulated plastic strain, p, corresponds to the hydrostatic stress, R, equals the gas constant and, T, the temperature. The viscosity function is determined by Eyring properties (A_0_, H, μ, τ_0_, a constant, the activation energy, a pressure coefficient and characteristic stress, respectively [27]) and the softening properties (h, D_∞_, these being the saturation value and the hardening parameter), respectively. The former determines the yielding behaviour of the material, and the latter determines the post-yielding softening behavior [29].

The meshes of the four geometries are shown in Figure 3b. General contact interfaces between the lattice structure and the loading plates were defined in the finite element model. Here, a mesh sensitivity analysis was undertaken by varying the mesh and the density of the mesh for the various scaled models. General contact interfaces between the lattice specimen and the loading plates were defined in the finite element model. 

The properties for the finite element model were generated from the stress–strain traces for each scale size. The Poisson’s ratio in this study was taken as ν = 0.3 [30]. The Eyring (yielding) properties were determined by the fitting yield point at different strain rates. A curve-fitting procedure was conducted to obtain the Eyring (yield) parameters in order to minimize the difference between FE prediction of and experimental data. The softening and hardening parameters including the post-yield regime, h, the saturation value, D_∞_, and the hardening parameter, H, were obtained from the test data [29]. The material parameters for the constitutive model listed in Table 3 [29,30,31].

## 4. Results and Discussion

### 4.1. Surface Characteristics

An initial study investigated the surface characteristics of the 3D-printed parts. Figure 4 shows polished cross-sections of struts taken from the four scaled sizes of the lattice structure. All four micrographs clearly highlight the uneven nature of the surfaces in these samples, reflecting the roughly circular nature of the PLA filament used in their manufacture. Clearly, the level of surface unevenness is similar in all four sizes of the sample, suggesting that this contoured surface profile is likely to have a greater impact on the smallest samples, where the relatively high level of surface unevenness is likely to more greatly affect the measured mechanical properties. The micrographs also highlight the presence of voids within the cross-sections, defects that ranged in size up to approximately 200 microns. Once again, the presence of these defects is likely to have a greater impact on the smaller samples, given that they will occupy a larger percentage of the cross-sectional area.

### 4.2. Compression Tests on the Scaled Solid Cubes

Figure 5a shows typical load–displacement traces following compression tests on the four sizes of scaled solid cube. All of the traces exhibit similar trends, with the force increasing rapidly to a peak value before reaching a relatively constant plateau value that extends for much of the testing regime. During the final stages of the test, the force begins to rise as the sample flattens and begins to be crushed between the plattens. Following testing, the load–displacement traces were normalized by dividing the force by the square of the scale size and the displacement by the scale size. The resulting normalized traces are shown in Figure 5b. An examination of the figure suggests that all four traces collapse onto a reasonably unique curve. The evidence suggests that the initial response of the ¼ scale sample differs from that of the three remaining scale sizes. It is worth noting, once more, that the upper and lower surfaces of this very small sample contained geometric irregularities, most notably small ridges aligned in the print direction. During the initial stages of the test, these irregularities were flattened, resulting in a lower initial slope, as evidenced in the trace. There are also small differences in the traces following the yield point in curves, with the force dropping slightly in the two larger sizes and increasing slightly in the two smaller sizes.

Figure 6 shows the variation in compression strength with scale size. From the figure, it is evident that the compressive strength of the sample increases slightly with sample size, passing from 64.2 MPa for the ¼ size sample to 75.9 MPa when n = 1, representing an 18.2% increase in strength. An examination of the samples after testing indicated that plastic deformation and barreling were the predominant failure mechanisms at all scale sizes. There were no visible differences in the appearance of the four sample sizes.

### 4.3. Flexural Tests on the Scaled Sizes of Rectangular Samples

Figure 7a shows typical load–displacement traces following flexural tests on the four scaled sizes of the PLA specimen. All four traces are relatively similar in appearance, with the load increasing in a non-linear fashion to a maximum value before decreasing rapidly when the sample fractures into two parts directly under the central loading pin. As before, the force was normalised by n^2^ and the displacement by n to yield the scaled traces shown in Figure 7b. Initially, all four traces collapse onto a single line, highlighting the scalability of the elastic properties. The traces begin to diverge in the non-linear region, although it is not possible to identify any clear trends linked to scale size. Given the significant differences in the sizes of the four samples (the volume of the largest is 64 times that of the smallest), the level of agreement is generally considered to be good. In each case, the maximum value of force was used to determine the flexural strength of the polymer samples and these values are summarized for the four scaled sizes in Figure 8. An examination of the figure indicates that the flexural strength increases as the sample size increases, passing from 70 MPa to 82.5 MPa, in going from the ¼ scale size to n = 1. This represents a 17.9% increase in strength over this range of scale sizes. It is worth noting that the thickness, t, of the ¼ scale samples was 2.3 mm and the approximate peak to trough distance or the surface rugosity was 100 microns, meaning that regions of the sample may be 200 microns thinner. Given that the stress has a 1/t^2^ dependency, this surface roughness effect is likely to have a much greater impact on the smaller samples. Similarly, voids with dimensions of up to 200 microns are likely to have a much greater influence on the mechanical performance of the smallest samples.

### 4.4. The Influence of Build Direction on the Properties of the Lattice Structures

Initial attention in this part of the study focused on the effect of build direction on the compression properties of the lattice structures. This is likely to be an important build parameter given that failure may involve different failure modes associated with complex stress fields in the joint regions. Gautam et al. [32] investigated the compressive properties of Kagome truss unit cell structures based on ABS and showed that the build angle influenced the average peak strength and effective stiffness due to differences in the strut dimensions as well as the anisotropic response of the truss structures. Here, full-size (i.e., n = 1) lattice structures were manufactured with the build direction either parallel (referred to as horizontal in the figure) to the face plates or transverse (termed vertical) to them. Figure 9 shows load–displacement traces corresponding to samples built in the two directions. Interestingly, both types of sample exhibit very similar load–displacement traces, with the load increasing to an initial peak before dropping rapidly as the central members of the lattice begin to deform at their joints as they move laterally outwards. The load then begins to increase once more as the upper and lower portions of the structure are subjected to increasing loads. The failure processes in the vertically and horizontally manufactured lattice structures are shown in Figure 10. Figure 10a shows the failure processes in the lattice manufactured in the horizonatal direction, i.e., the build direction is from the bottom to the top in the figure. As the sample is loaded, the joints linking the centermost members begin to deform and fracture. During this stage of the test, the upper and lower members remain intact. This phase of loading coincides with the drop in loading that was observed in the load–displacement trace. Further loading results in the central member being forced togther, Figure 10a(iii). During the final stages of the test, the upper and lower parts of the structure are loaded, leading to the deformation and fracture of these members within this lattice. Similar failure mechanisms can be observed during compression loading of the vertically manufactured lattice, Figure 10b, where failure again initiates at the mid-depth, before extending to the upper and lower regions of the sample. A comparison of Figure 10a, b highlight the similarties between the two samples, observations that agree with the trends in the load–displacement traces in Figure 9. This information suggests that the direction of build does not have a significant affect on the load-bearing properties of these lattice structures.

### 4.5. Compression Tests on the Four Scaled Sizes of Lattice Structure

Figure 11a shows load–displacement traces following compression tests on the four sizes of lattice structure. All of the curves exhibit an initial linear region with the force increasing to a maximum value before declining steadily to a much lower value. The traces then reach a minimum, after which the load starts to increase once more, typically to a value that is higher than the initial peak. Figure 11b shows the resulting normalised traces, where the load and displacement have been scaled using the procedures reported earlier. Clearly, all of the traces are reasonably similar in appearance, although there are clear variations in the maximum load values. All traces exhibit a similar initial stiffness and a similar rate of decline in the post-maximum force. Interestingly, the force begins to increase once more at a similarly scaled displacement, approximately 25 mm.

Figure 12 shows the failure modes in the four sizes of lattice structure. An examination of the figure indicates that the fracture processes are similar in each case, with initial failure occurring in the central region of the sample, typically as a result of a bending failure at edges of the centermost cell. Closer examination highlights cracks and small holes in these regions at intermediate levels of deformation. This central region then collapses in on itself, before the lowermost struts start to deform and collapse. This evidence suggests that scaled replicas can be used to predict the failure mode in larger structures with a high degree of confidence. This is an important observation, given that the mass of the largest sample is sixty-four times that of the n = 1/4 lattice. This suggests that significant time and cost savings could be realized by employing scaled replicas to identify failure modes in larger components.

Figure 13 summarizes the compression strengths of the four scaled sizes of lattice structure. Here, it is clear that the compression strength increases rapidly with increasing scale size. This is most clearly highlighted by comparing the n = ¼ and n = 1 samples, where the latter is sixty percent stronger that the former. Clearly, this difference in properties is significantly greater than that observed in the compression and flexural tests and is likely to be linked to the relative influence of the manufacturing defects, such as the aforementioned voids, particularly in the joint regions, in the smaller samples. Further, given that the dimension of the minor axis in the ¼ scale sample was just 2.25 mm, surface roughness effects amounting to up to 0.2 mm are likely to have a significant impact on the bending response of the struts.

### 4.6. Numerical Results

Figure 14a compares the predicted and experimental force–displacement traces for the compression tests on the PLA cubes. A comparison of the various traces in the figure indicates that the model accurately predicts the compressive response of the polymer over the four sizes of scaled sample. From the figure, it is clear that the model captures the initial elastic response, the yield characteristics, as well as the post-yield response.

Figure 14b compares the predictions of the FE model with the experimental load–displacement traces following the tests on the four scaled sizes of lattice structure. In the model, the mechanical properties for each scale size were based on those measured on the particular scale size during mechanical testing. A comparison of the numerical and experimental data suggests that the FE model predicts the experimental data with a reasonable level of success. In general, the models over-estimate the experimental data, probably due to the fact that the model does not account for the presence of defects with the cross-section as well as assuming a smooth constant diameter for the various struts within each sample. All FE traces predict the steady decrease in load after the initial peak, as well as the subsequent increase during the middle stages of the test. The resulting deformation modes that are predicted in the n = 1 lattice sample are compared with the experimentally observed failure mechanisms in Figure 15. A comparison of the two sets of figures shows that the FE model captures the failure processes in the sample with reasonable success. Failure develops with the central region compressing as the struts push outwards. This compressing motion causes the struts to fracture in the test samples, although this is not evident in the FE model. Continued loading leads to the more rigid upper and lower regions deforming and collapsing as the lattice enters the densification stage and the force begins to rise rapidly. Similar deformation mechanisms were observed in the remaining three scaled sizes.

## 5. Conclusions

Scaling effects in the mechanical properties of a range of structures manufactured from a PLA polymer using additive manufacturing techniques have been investigated. Initial tests focused on investigating the size effects on the compression properties of cube-shaped samples, where the edge length of the cube was scaled to give nominally identical sample sizes. Here, it was shown that the compression strength increased by approximately 18% as the edge length was increased by a factor of four. A similar increase was observed in the flexural strength of the polymer as the length, width and thickness dimensions of the beams were increased by a similar scale factor. It is believed that surface roughness effects and the relative size of defects play a more prominent role in determining the strength of the smaller samples. In the second part of this investigation, compression tests were undertaken on lattice structures based on the body-centered cubic design. Here, four identically scaled lattice geometries were tested at appropriately scaled crosshead displacement rates. Although similar failure modes were observed in the four sizes of samples, significant differences in compression strength were observed with values increasing by sixty percent as the scale size was increased from n = 1/4 to n = 1.

Finally, the compression properties of the lattice structure were modelled using finite element analysis techniques. Although the FEA models correctly predicted the failure modes that were observed in the lattice structures, they tended to over-estimate their strength characteristics, probably due to the fact that the struts within the lattices were assumed to be smooth and defect-free. The evidence presented here indicates that scaling effects do occur in additively manufactured structures and that small-scale prototypes are likely to present a conservative estimate of the load-bearing characteristics of full-scale structures.

## Figures and Tables

**Figure 1 polymers-13-03967-f001:**
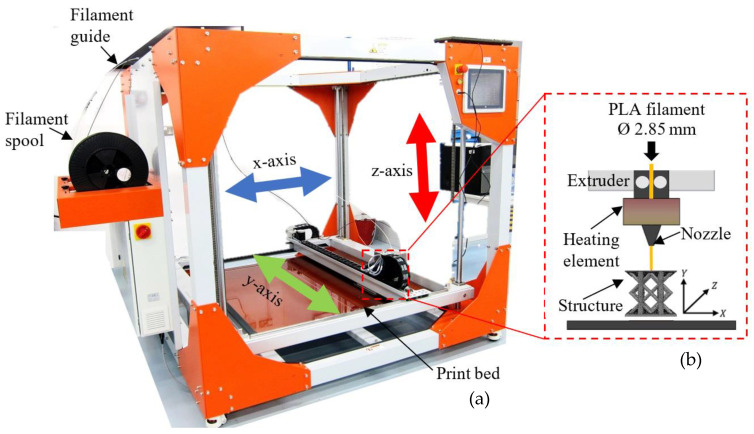
(**a**) Photograph of the BigRep ONE printer and (**b**) a schematic diagram of the three-dimensional printing process.

**Figure 2 polymers-13-03967-f002:**
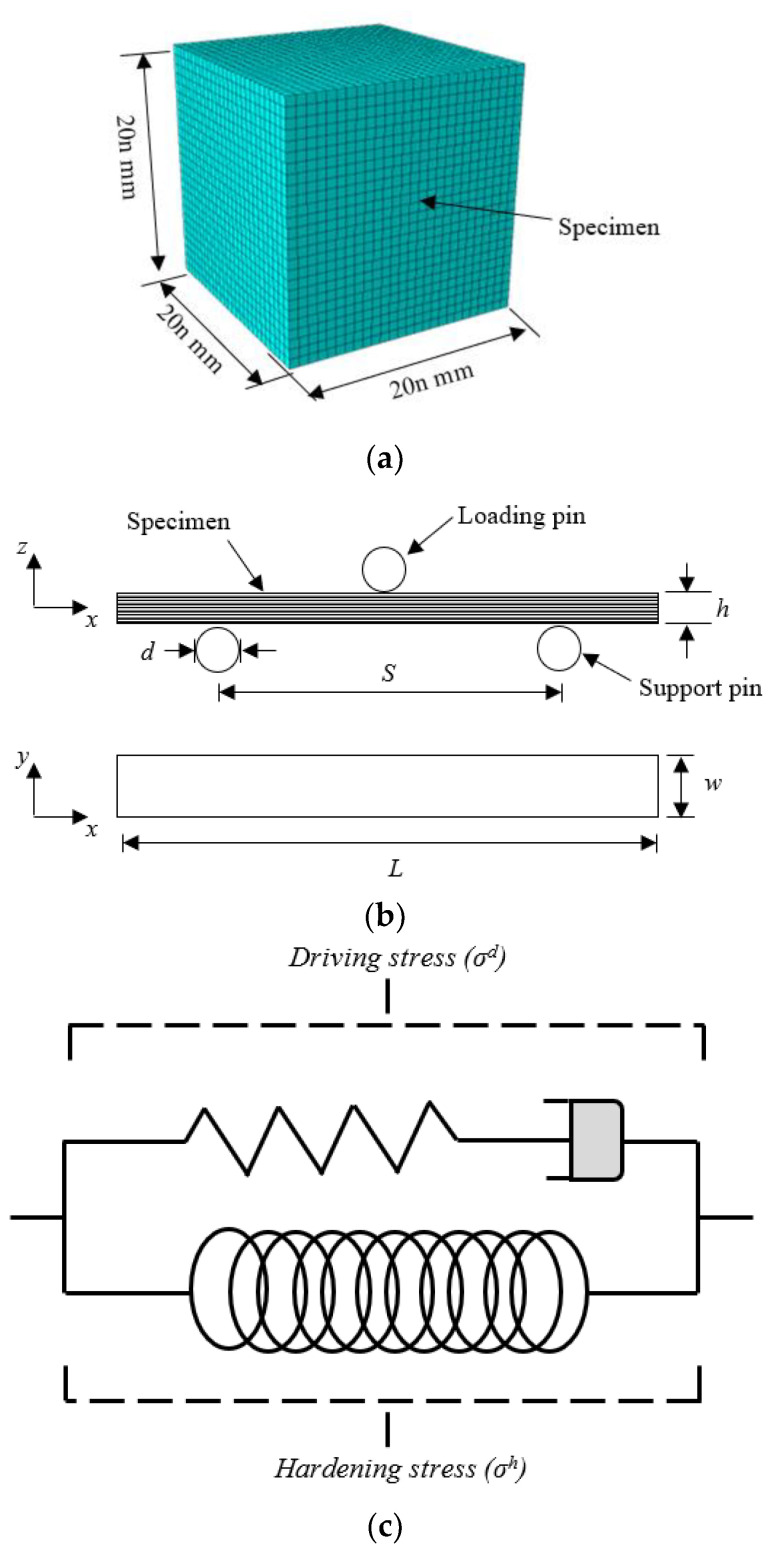
(**a**) Schematic of the cube test sample with an edge length of 20n mm. (**b**) Schematic of the flexural test sample with edge lengths of 200n × 40n × 8n mm^3^. (**c**) The viscoelastic model.

**Figure 3 polymers-13-03967-f003:**
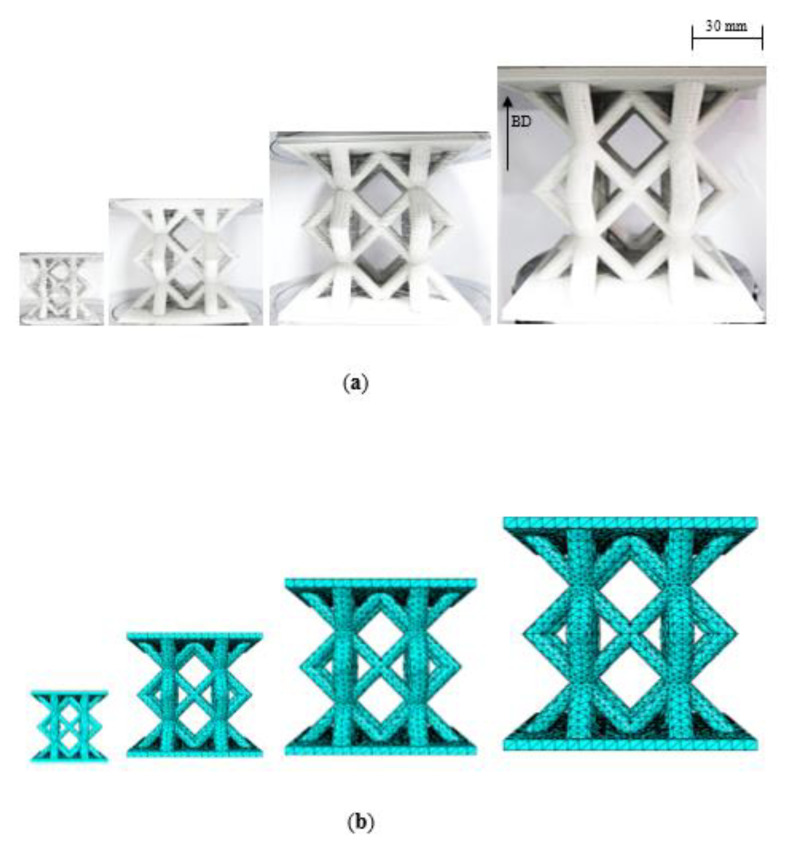
(**a**) Photographs of the four scaled lattice samples where the build direction (BD) is indicated by an arrow. (**b**) Finite element models of the four scaled sizes lattice structures.

**Figure 4 polymers-13-03967-f004:**
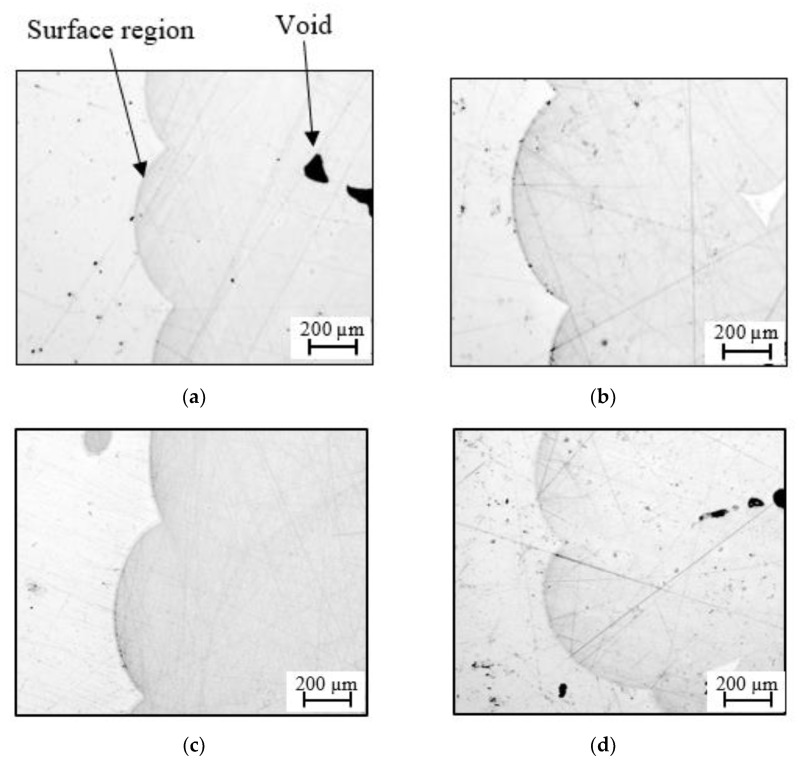
Cross-sections of the surface regions of the four scaled sizes of lattice structure (**a**) n = 1, (**b**) n = 3/4, (**c**) n = 1/2 and (**d**) n = 1/4.

**Figure 5 polymers-13-03967-f005:**
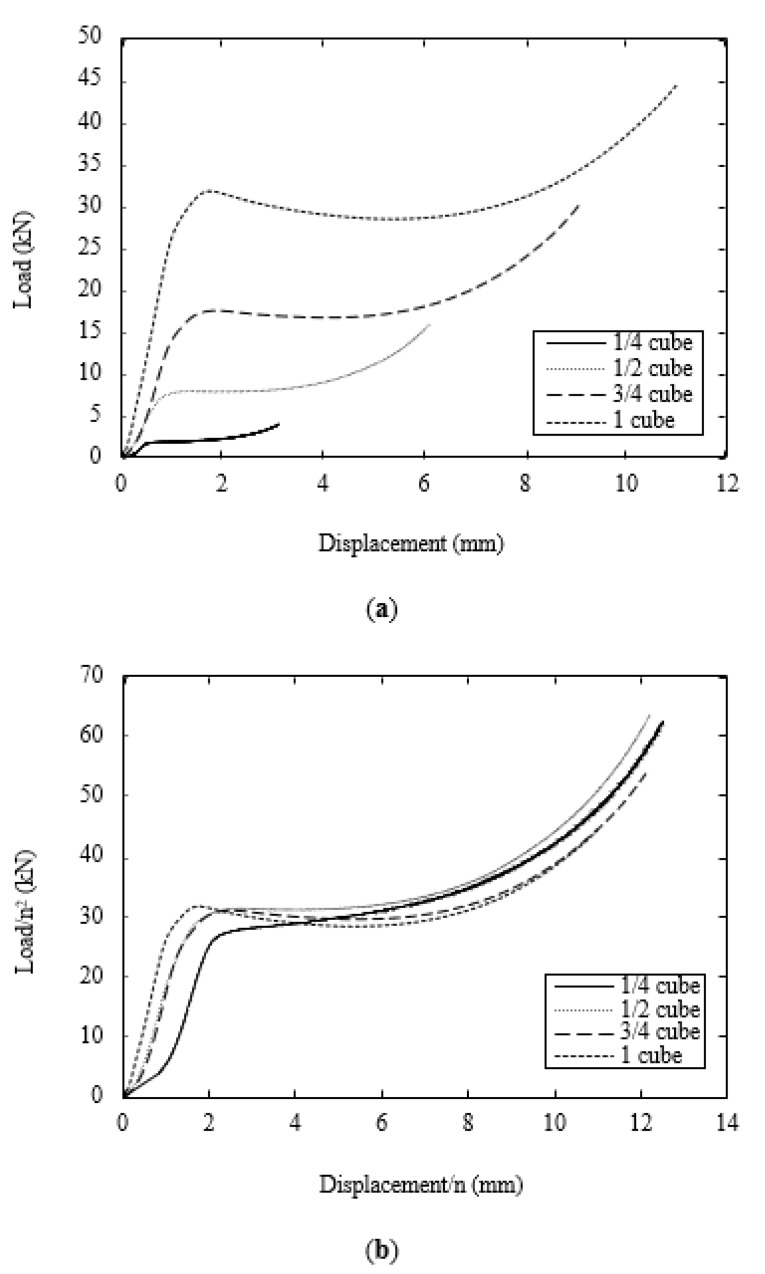
(**a**) Typical load–displacements traces following compression tests on the cubes. (**b**) Typical scaled load–displacements traces following compression tests on the cubes.

**Figure 6 polymers-13-03967-f006:**
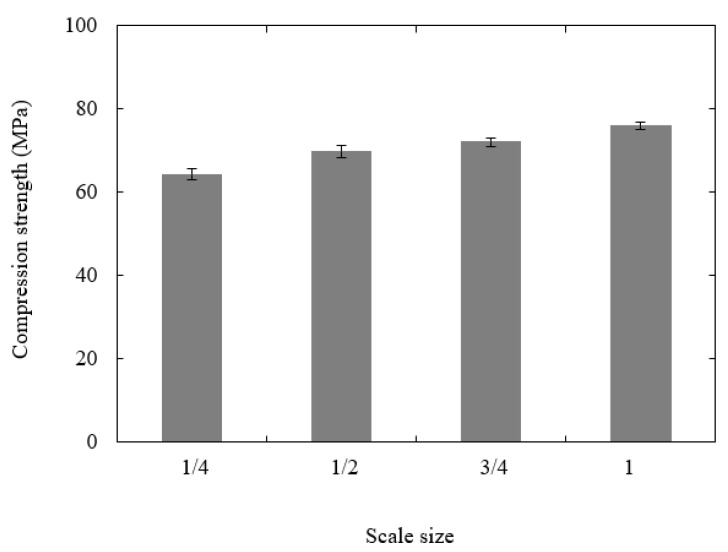
Plot of compression strength vs. scale size for the PLA cubes.

**Figure 7 polymers-13-03967-f007:**
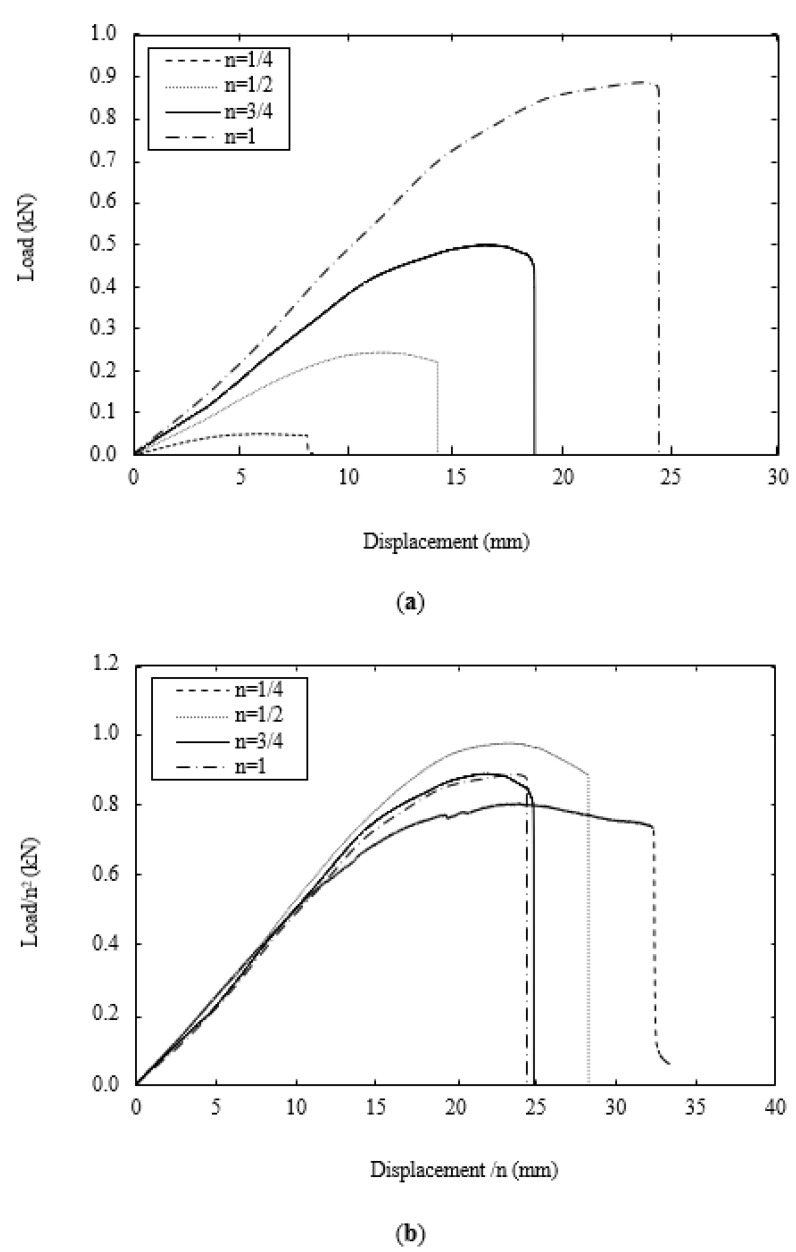
(**a**) Typical load–displacements traces following flexural tests on the rectangular samples. (**b**) Typical scaled load–displacements traces following flexural tests on the rectangular samples.

**Figure 8 polymers-13-03967-f008:**
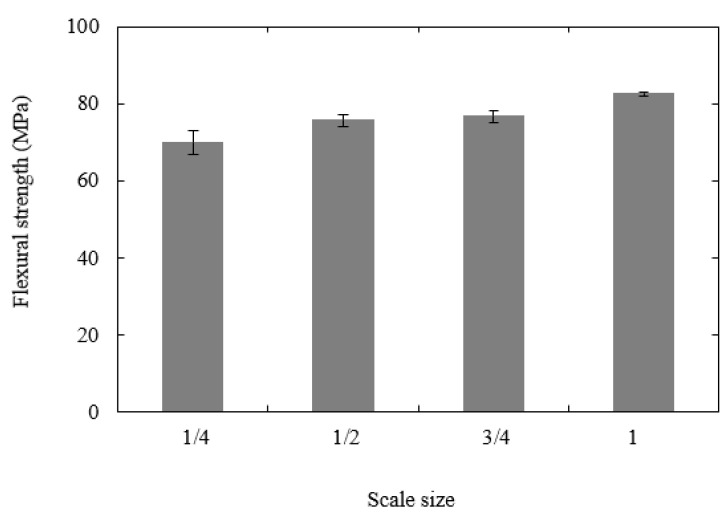
Plot of flexural strength vs. scale size.

**Figure 9 polymers-13-03967-f009:**
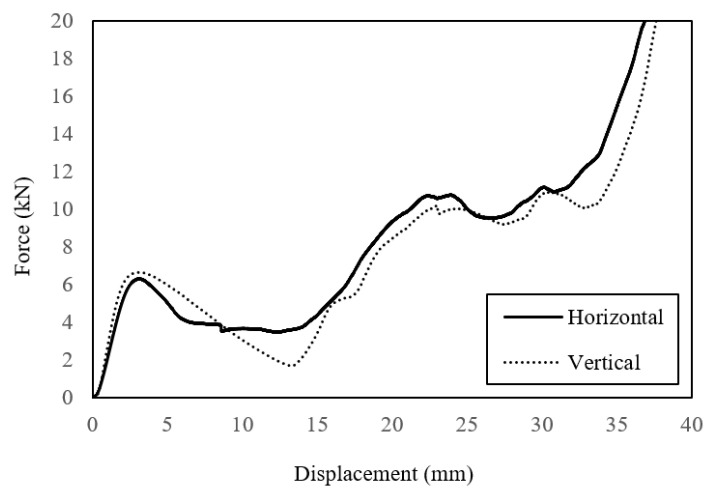
Load–displacement traces following compression tests on lattice structures (n = 1) built vertically and horizontally.

**Figure 10 polymers-13-03967-f010:**
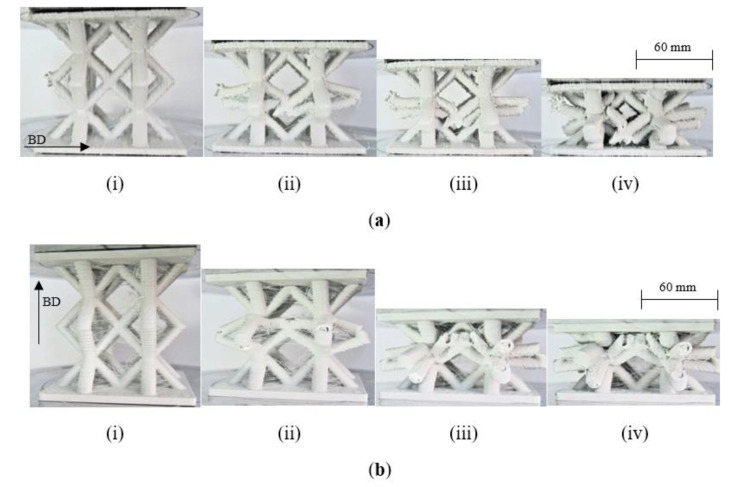
Photographs showing the effect of build direction on failure mechanisms in a lattice structure. The build direction is indicated by the arrow. (**a**) Lattice structure (n = 1) manufactured in the horizontal direction and (**b**) lattice structure (n = 1) manufactured in the vertical direction.

**Figure 11 polymers-13-03967-f011:**
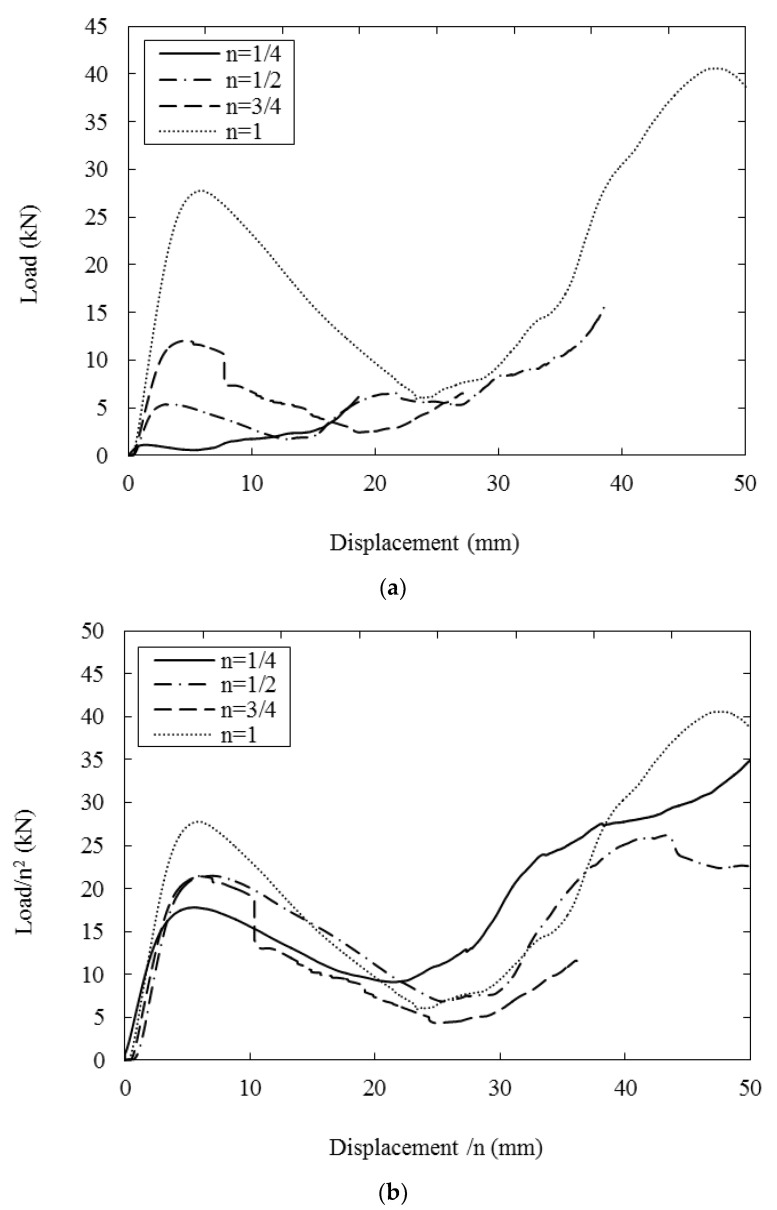
(**a**) Typical load–displacements traces following tests on the four scaled sizes of lattice structure. (**b**) Typical scaled load–displacements traces following tests on the four scaled sizes of lattice structure.

**Figure 12 polymers-13-03967-f012:**
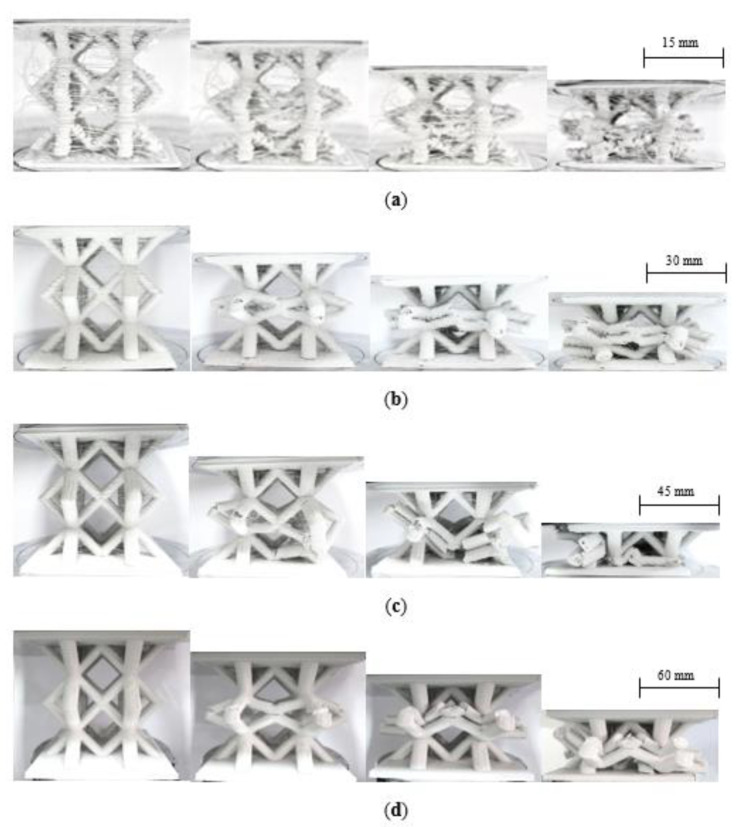
Photographs of the four scaled sizes of lattice structures under compression. (**a**) n = ¼, (**b**) n = ½, (**c**) n = ¾ (**d**) n = 1.

**Figure 13 polymers-13-03967-f013:**
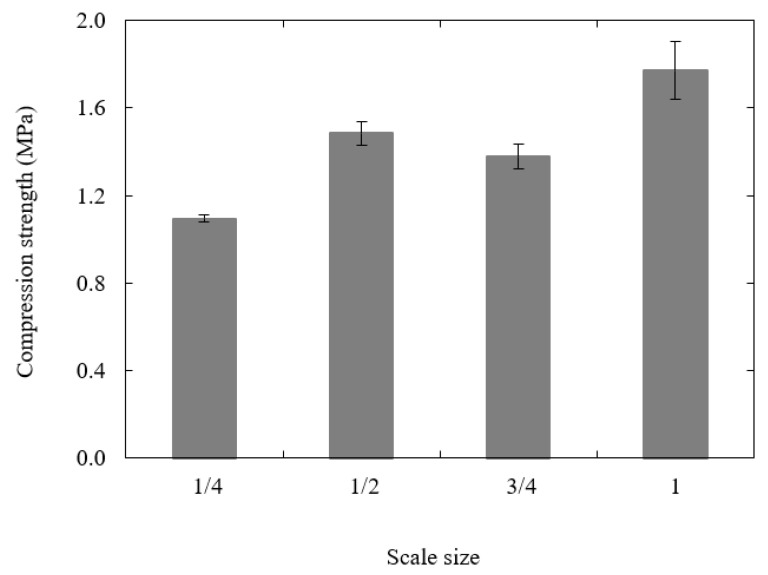
Plot of compression strength of the lattice structures vs. scale size.

**Figure 14 polymers-13-03967-f014:**
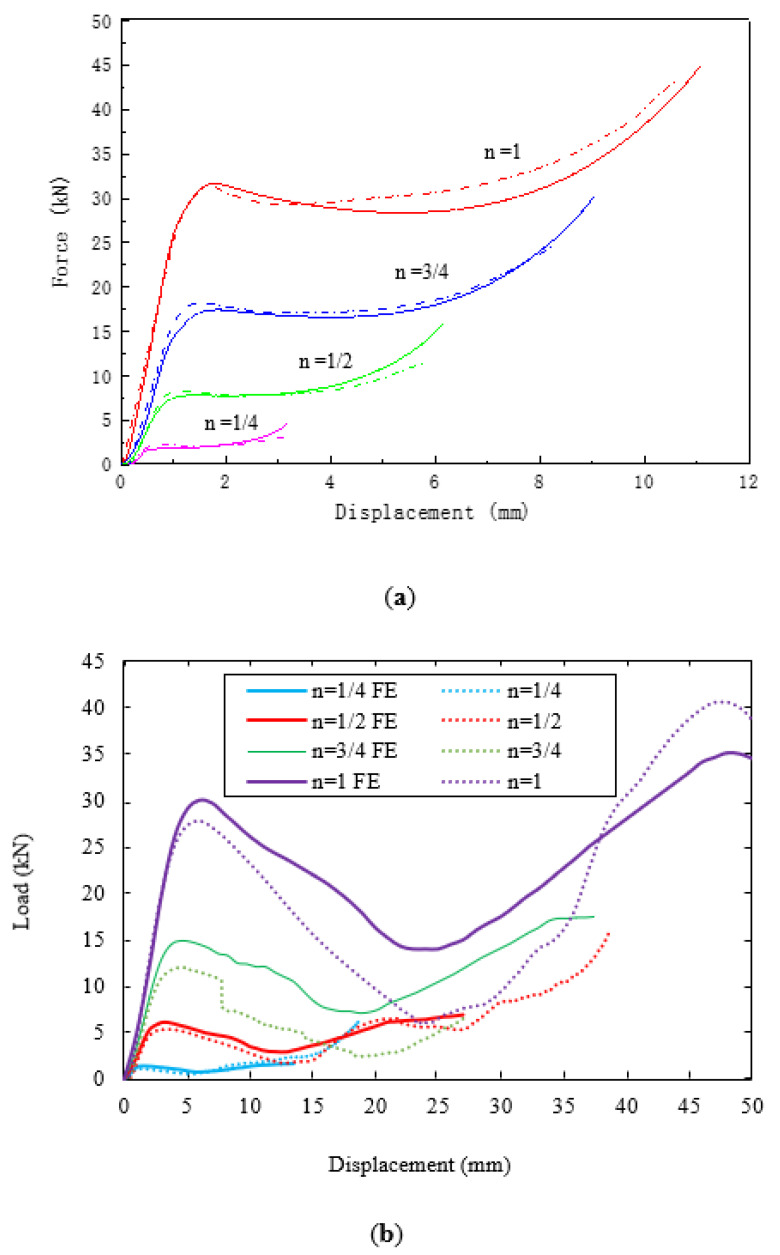
Comparison of the predicted (dashed lines) and experimental (solid lines) force–displacement traces following compression tests on (**a**) the four scaled sizes PLA cubes and (**b**) the lattice structures.

**Figure 15 polymers-13-03967-f015:**
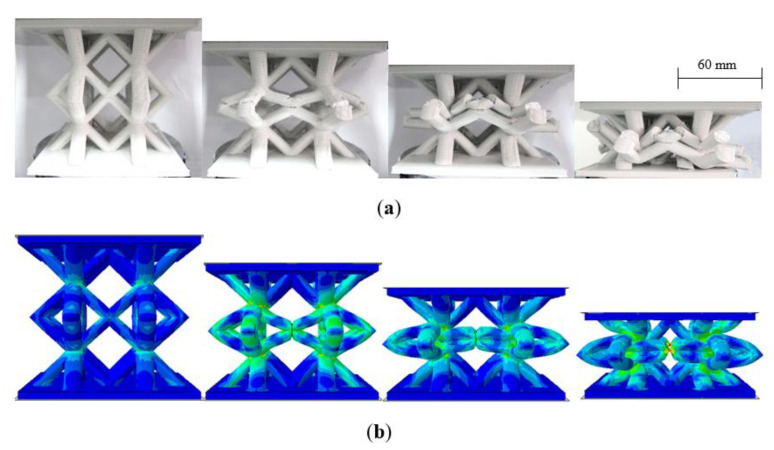
Comparison of (**a**) the experimentally observed and (**b**) the predicted failure modes in the largest (n = 1) lattice structure.

**Table 1 polymers-13-03967-t001:** Parameters employed for manufacturing the test samples.

Parameter	Details
Filament Diameter	2.85 mm
Printing Nozzle Temperature	205 °C
Printing Bed Temperature	60 °C
Nozzle Diameter	0.6 mm
Layer Thickness	0.5 mm
Default Printing Speed	3000 mm/min.
Infill Percentage	100%

**Table 2 polymers-13-03967-t002:** Average dimensions and test conditions for the compression, flexural test and lattice samples.

Parameter	n	Sample		
		Compression	Flexural Test	Lattice
Length (mm)	¼	5.3	50	31.8
	½	10.5	100	61.5
	¾	15.6	150	91.7
	1	20.5	200	120.2
Width (mm)	¼	5.3	9.6	31.9
	½	10.5	19.8	61.5
	¾	15.6	30.0	91.7
	1	20.5	39.6	120.2
Height (mm)	¼	5.3	2.3	27.4
	½	10.5	4.7	55.3
	¾	15.6	6.6	82.1
	1	20.5	8.6	107.1
Crosshead displacement rate (mm/min)	¼	1	2	2
	½	2	4	4
	¾	3	6	6
	1	4	8	8

**Table 3 polymers-13-03967-t003:** Material properties of PLA used in numerical analyses [29,30,31].

Material Properties of PLA	
Young’s modulus in the elastic regime, *E* (MPa)	1136
Poisson’s ratio, ν	0.3
Density, ρ (g/m^3^)	1250
The absolute temperature, T (k)	296.15 (room temperature)
The universal gas constant, R (J/mol·K)	8.3143
The Eyring (yielding) properties/parameters	
The scalar /preexponential factor, A_0_ (s)	3.65 E-32
The activation energy, ΔH (J/mol)	2.062 E+5
A characteristic stress, σ_0_ (MPa)	5.7
Pressure coefficient, μ	0.3
The softening parameters	
The softening properties, h	450
The saturation value of softening, D_∞_	2.16
Hardening parameter, H (MPa)	10

## Data Availability

The data presented in this study are available on request from the corresponding author.

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
