# Peer review of "Geometrical Scaling Effects in the Mechanical Properties of 3D-Printed Body-Centered Cubic (BCC) Lattice Structures"

_polymers, 2021, doi:10.3390/polym13223967_

Round 1

Reviewer 1 Report

This article investigated mechanical properties focusing on the dimensional scaling effect of 3D printed polylactic acid (PLA) based lattice structure.

The investigation is systematic and provides a good and large experimental result also supported by numerical analysis. Hence it is worth considering in POLYMERS. However, the following concerns are to be addressed before acceptance.

Please consider modifying the word “scaling” in the title, it could be misleading. Here it deals with geometrical or dimensional scaling, I suggest making it clearer to readers. Also, could the author include the name of lattice in the title ( also see my later comment about it)

The abstract is too long and two paragraphed, shorten it into a single paragraph

Line 13, ..200n, 4 and 8n… is it a typo, n is missing after 4?

Line 35 …please consider replacing the word “dramatic” - possible alternative “unprecedented”

Line 97…do not start a sentence with “While”

Line 109, “Experimental procedure” replace with “Materials and method”

Figure 1 is too simplistic; it is just a photograph. Please convert it into a scheme, this Figure should tell the whole story about what the article is going to deliver.

Can you please combine table 2, 3 and 4 à simply provide information about compression, flexural and lattice sample at 3 different columns

Line 206, replace word “procedure” by method or analysis

I do not think Table 5, which is taken from other studies, needs to be provided here. References should be enough. It is just increasing the length of the article

What is the purpose of Figure 4? I do not see any significance or provide better images using SEM and point by arrows or highlight where to focus. Now it is a simple collection of some random images

Figure 10: the lattice looks like a Kagome structure? Could authors compare to the following article and comment what is the difference or novelty between your and their lattice? Also, compare the compressive/failure properties between such lattices.

https://www.sciencedirect.com/science/article/abs/pii/S0264127517309437

There are a large number of figures, please make multi-panel figures for a similar type of results to reduce the number as well as to increase the excellence of the work. Also, note that there are largely visible gaps in the Figures, please make Figure more compact.

Please also cite these articles focused on polymer printing

https://www.sciencedirect.com/science/article/pii/S0264127519308524

https://www.mdpi.com/2073-4360/13/18/3101

Reviewer 2 Report

The manuscript entitled "Scaling Effects in the Mechanical Properties of 3D Printed Lattice Structures" by Aziz et al. described the effects of size in the mechanical properties of 3D printed PLA based lattice structures. Overall, the study design, experimental details, results and presentation were impressive.
